# Quantitative Comparison of Statistical Methods for Analyzing Human Metabolomics Data

**DOI:** 10.3390/metabo12060519

**Published:** 2022-06-04

**Authors:** Mir Henglin, Brian L. Claggett, Joseph Antonelli, Mona Alotaibi, Gino Alberto Magalang, Jeramie D. Watrous, Kim A. Lagerborg, Gavin Ovsak, Gabriel Musso, Olga V. Demler, Ramachandran S. Vasan, Martin G. Larson, Mohit Jain, Susan Cheng

**Affiliations:** 1Cardiovascular Division, Brigham and Women’s Hospital, Harvard Medical School, Boston, MA 02115, USA; mir.henglin@cshs.org (M.H.); bclaggett@bwh.harvard.edu (B.L.C.); jantonelli@ufl.edu (J.A.); ovsak.gavin@gmail.com (G.O.); gabe@biosymetrics.com (G.M.); 2Smidt Heart Institute, Cedars-Sinai Medical Center, Los Angeles, CA 90048, USA; ginoalberto.magalang@cshs.org; 3Department of Statistics, University of Florida, Gainesville, FL 32603, USA; 4Department of Biostatistics, Harvard T.H. Chan School of Public Health, Boston, MA 02115, USA; 5Departments of Medicine & Pharmacology, University of California San Diego, La Jolla, CA 92161, USA; m1alotaibi@health.ucsd.edu (M.A.); jeramie.watrous@gmail.com (J.D.W.); kimalehmann@gmail.com (K.A.L.); 6BioSymetrics Inc., Huntington, NY 11743, USA; 7Preventive Medicine, Brigham and Women’s Hospital, Harvard Medical School, Boston, MA 02115, USA; odemler@bwh.harvard.edu; 8Preventive Medicine, Department of Medicine, Boston University Medical Center, Boston, MA 02118, USA; vasan@bu.edu; 9Framingham Heart Study, Framingham, MA 01702, USA; mlarson@bu.edu; 10Biostatistics Department, School of Public Health, Boston University, Boston, MA 02118, USA

**Keywords:** metabolomics, statistical methods, univariate, multivariate

## Abstract

Emerging technologies now allow for mass spectrometry-based profiling of thousands of small molecule metabolites (‘metabolomics’) in an increasing number of biosamples. While offering great promise for insight into the pathogenesis of human disease, standard approaches have not yet been established for statistically analyzing increasingly complex, high-dimensional human metabolomics data in relation to clinical phenotypes, including disease outcomes. To determine optimal approaches for analysis, we formally compare traditional and newer statistical learning methods across a range of metabolomics dataset types. In simulated and experimental metabolomics data derived from large population-based human cohorts, we observe that with an increasing number of study subjects, univariate compared to multivariate methods result in an apparently higher false discovery rate as represented by substantial correlation between metabolites directly associated with the outcome and metabolites not associated with the outcome. Although the higher frequency of such associations would not be considered false in the strict statistical sense, it may be considered biologically less informative. In scenarios wherein the number of assayed metabolites increases, as in measures of nontargeted versus targeted metabolomics, multivariate methods performed especially favorably across a range of statistical operating characteristics. In nontargeted metabolomics datasets that included thousands of metabolite measures, sparse multivariate models demonstrated greater selectivity and lower potential for spurious relationships. When the number of metabolites was similar to or exceeded the number of study subjects, as is common with nontargeted metabolomics analysis of relatively small cohorts, sparse multivariate models exhibited the most-robust statistical power with more consistent results. These findings have important implications for metabolomics analysis in human disease.

## 1. Introduction

Mass spectrometry-based measurements of small molecule metabolites, also known as metabolomics, has emerged as a powerful tool for phenotyping biochemical variation in health and disease across organisms. Accordingly, there has been a rapidly growing interest in applying metabolomics to clinical studies of human disease traits [1,2]. Robust statistical methods are particularly needed to examine associations between metabolites detected in peripheral blood circulation with disease traits in humans; in this context, false discovery remains a key concern for clinical biomarker studies [3,4,5]. The statistical analysis challenges posed by human metabolomics data arise from multiple sources. For instance, metabolomics data collected from a given biospecimen represents metabolite variation at a particular point in time and in a particular context: whereas a portion of the variability reflects the relatively stable components of the organismal metabolome, another component reflects the dynamic portion of the metabolome that varies substantially over time and in response to a number of exposures. Such mixed structures can lead to a high degree of variation for a given metabolite level across individuals. Additionally, due to common pathways of enzymatic production or exposures of origin, metabolites can demonstrate a high degree of intercorrelation, and this intercorrelation may vary between individuals or subgroups depending on disease state, exposures, or other factors.

Initial clinical studies involving targeted metabolomics approaches have used relatively conservative statistical approaches to analyze up to 200 variables, such as Bonferroni correction of multiple t-tests or the false discovery rate (FDR) [6]. Additionally, methods of accounting for multiple hypothesis testing have similarly assigned more or less conservative thresholds for defining statistical significance. In the absence of considering intercorrelations between individual metabolites at the outset, data analyses will tend to favor identifying metabolites from a singular biological pathway, with secondary or tertiary associations (potentially representing important orthogonal pathways) being forced to reach lower levels of statistical significance based on rank ordering alone. For this reason, traditional statistical approaches are believed to offer limited sensitivity for high-dimensional metabolomics analyses. Thus, several alternative methods have been proposed to more effectively select metabolites associated with a given outcome [7,8,9,10,11]. These methods have begun to surface from analyses of other molecular phenomics datasets [9,10,12,13], although they may differ in structure relative to metabolomics datasets. Each statistical method has intrinsic strengths and weaknesses, and the extent to which they may be more or less suited for a given metabolomics analysis is not known, but is likely to depend on number of metabolites assayed, sample size, and frequency or type of clinical outcome. Therefore, we sought to formally test currently available statistical methods across a range of dataset types. By simulating clinical studies to test different outcome-based hypotheses and validating findings using real metabolomics data, we aimed to assess the suitability of statistical methods for the analysis of metabolomics data across a range of clinical data settings. Our investigation primarily focused on studies of moderate to large sample size that are predominantly observational in design, wherein the primary exposures are expected to have modest to moderate associations with a given outcome, rather than large magnitudes of effect as might be seen in interventional studies.

## 2. Results

### 2.1. Statistical Analyses of Simulated Metabolomics Data

Metabolomics studies of human samples can vary substantially by sample size, the number of metabolites assayed, and the type and frequency of a clinical outcome of interest, with each of these factors potentially influencing statistical analysis results. To evaluate statistical methods for handling a variety of datasets, metabolomics data were simulated for clinical studies of a varying number of study subjects, number of metabolites, and outcome type (continuous vs. binary). A total of six traditional statistical (Bonferroni, FDR, and principal component regression (PCR)), and statistical learning (least absolute shrinkage and selection operator (LASSO), sparse partial least squares (SPLS), and random forest) methods were used to analyze 1000 simulated metabolomics datasets (Figure 1 and Figure 2), with each evaluated for the likelihood of a metabolite being correctly identified as one of the top 10 most important metabolites with respect to a given outcome. For a simulated continuous outcome (Figure 1), all approaches performed similarly well, with the exception of scenarios with a large number of metabolites (M = 2000) or a small number of subjects (N = 200). At these extremes, multivariate approaches based on sparsity, LASSO, and SPLS outperformed univariate approaches. In the case of a binary outcome (Figure 2), optimal statistical methods were less apparent. Univariate approaches based on the linear model performed slightly better than multivariate approaches with small sample sizes. As the sample size increased, results approximated those observed in the continuous case, where sparse multivariate methods such as sparse partial least squares discriminant analysis (SPLSDA) outperformed the other approaches (Figure 2). In secondary analyses, we repeated the simulation experiments to include smaller sample sizes (i.e., N = 50 and N = 100) that may be used for some studies despite acknowledged limitations in statistical power. In these analyses, we observed the expected trend of generally reduced sensitivity for smaller sample sizes, while specificity was generally preserved, corresponding to a greater positive predictive value (PPV) in smaller sample sizes. The exception was SPLS models, for which false positive rates increased in the smallest samples sizes along, corresponding to reduced PPV (Appendix A).

An equally important aspect of a statistical procedure is the identification of important metabolites via variable selection or significance testing. Variable selection is not generally possible with PCR and random forest analyses, precluding assessment of these approaches for prioritizing individual metabolites. In either the continuous or binary settings, univariate approaches performed worse as the number of study participants increased (Figure 1 and Figure 2). While counterintuitive given that statistical performance in general is enhanced with sample size, due to the frequently correlated nature of metabolomics data, false positives (i.e., variables identified as significantly associated but not pre-specified as directly associated with the clinical outcome) increased substantially with univariate methods as a result of these variables being selected due to their correlation with ‘true positive’ metabolites (i.e., variables prespecified as directly associated with the clinical outcome) (Figure 1 and Figure 2). This contributed to poor positive predictive value and reduced specificity for all of these approaches, both of which are important concerns for clinically relevant biomarker discovery. Multivariate approaches, by contrast, do not suffer this same drawback, as their performance improves as the sample size increases (Figure 1 and Figure 2). In the case of a continuous outcome, both LASSO and SPLS methods performed remarkably well, with SPLS slightly outperforming LASSO in terms of positive predictive value, negative predictive value, and number of false positives. An exception to this trend was seen in the smallest-sizes samples (i.e., N = 50 or N = 100), wherein the false positive rate was higher for SPLS than for LASSO models. Binary outcomes differed from statistical analysis of continuous outcomes due to different performance for the respective estimators at different sample sizes. In small sample sizes, univariate procedures with multiplicity correction had the best positive predictive value among all estimators (Figure 3). As the sample size increased to 1000 or 5000, multivariate approaches again outperformed univariate procedures, as both LASSO and SPLSDA obtained the highest positive predictive value and negative predictive value and the fewest false positives. Interestingly, for SPLSDA, the positive predictive value decreased from N = 1000 to N = 5000 as the number of false positives increased, although this was likely due to sensitivity of tuning parameter selection, which is required for the application of sparse methods.

Although we aggregated over all 10 significant metabolites to calculate measures of method performance, differences in operating characteristics such as power likely correspond with smaller effect sizes. While most methods identify associations with large magnitudes of effect, potentially important discrepancies can become apparent for associations with smaller effect sizes. Thus, we also examined variation in power across a range of effect sizes for different statistical approaches, and these results were concordant with those of aggregated data (Appendix A).

We also simulated scenarios that included negative as well as positive between-metabolite correlations in addition to pairs of highly intercorrelated metabolites representing molecular markers putatively derived from the same biological pathway. We observed that the results of these additional simulations were very similar to those produced by the primary simulations and reported herein (Appendix A), suggesting that our overall findings from the simulated data are relatively consistent across variations in simulated data structure. Collectively, these findings suggest the value of multivariate approaches for identifying metabolite markers that are associated with clinical traits.

### 2.2. Statistical Analyses of Experimentally Derived Metabolomics Data

Although the results reported above put forth a statistical framework for considering analysis of clinical metabolomics based on analyses of simulated data, we sought to compare our findings with those using actual “real world” experimentally derived metabolomics data. For these analyses, we used a nontargeted metabolomics-based panel of 1933 metabolites measured across 2895 individuals (see Methods). We restricted attention to the methods that would easily allow for individual variable (i.e., metabolite) importance selection in the dataset, which precluded random forest and PCR from entering into the analysis. Analyses using the three main statistical approaches (FDR, LASSO, and SPLS) revealed overlap (Figure 3) for only a minority of the total detected associations between metabolites and either a continuous variable (age) or a binary variable (sex).

We excluded from the Venn diagram the results from the Bonferroni correction, given it produces a subset of the same metabolites chosen using the less conservative FDR correction. We applied a false discovery rate of 0.1, which suggests that 10% of the metabolites on average should be false discoveries. For both outcomes, use of FDR resulted in a large number of statistically significant results, with >50% of all assayed metabolites (1281/1933 for age, 1312/1933 for sex) reaching the threshold, suggesting that an FDR correction of 0.1 was in fact detecting nearly all of the signals. The approaches rooted in sparsity, however, obtained solutions with far fewer metabolite “hits”. In both cases, LASSO analysis resulted in far fewer metabolites than FDR correction (206 for sex and 378 for age). By contrast, SPLS provided far fewer metabolite hits than either LASSO or FDR correction (93 for sex and 37 for age). In the case of both age and sex, SPLS did not identify any new metabolites beyond those found in the LASSO or FDR subsets. We found in this study that when implementing cross-validation to estimate tuning parameters of SPLS and SPLSDA, the cross-validation curve is relatively flat, an issue that has been previously encountered [8]. This suggests that different levels of sparsity were equally supported by the data, and we chose to use the sparsest option to identify the most important metabolites. SPLS is a relatively new approach, for which these issues have not been well addressed; this differs for the LASSO approach, wherein estimating the tuning parameter is straightforward using glmnet in R [14]. We repeated all analyses in the Framingham Heart Study dataset, while including an indicator variable for specimen batch in all models. As shown in Appendix A, results of these analyses were similar to those in the original analyses.

To provide further context for these results, we used a basic network analysis to visualize correlations between metabolite measures in the Framingham cohort and compared the relative location of metabolites identified in association with age or sex by the univariate and multivariate methods evaluated (Figure 4). We observed that univariate approaches tend to identify highly intercorrelated metabolites, whereas multivariate approaches exhibit a more parsimonious as well as broader selection of metabolites.

### 2.3. Results from Both Simulated and Experimentally Derived Data

The multiple statistical analysis approaches, when applied to both the simulated and the experimentally derived data, produced relatively comparable results with respect to very large numbers of metabolite markers identified by univariate compared to multivariate methods. Of the multivariate methods evaluated, LASSO appeared to perform slightly better than SPLS across the different types of simulated data structures, and especially those involving larger numbers of metabolites (Figure 1 and Figure 2). In comparison, SPLS appeared to be more selective when applied to the experimentally derived dataset (Figure 3). Given that selectivity alone is not necessarily a measure of true association, these results together suggest that results of either SPLS or LASSO would be reasonable to consider in a clinical study, particularly given that the metabolites SPLS identified as associated with either the continuous or binary outcomes overlapped with those identified by the univariate or alternate multivariate methods.

## 3. Discussion

Through extensive simulation, we investigated the relative merits of traditional statistical and statistical learning approaches for the analysis of human metabolite data. Using a data structure based on real-world metabolite data with varying sample size, metabolite number, and outcome measures, our results offer a framework for considering optimal statistical approaches for a given study. We found that penalized approaches favoring sparsity led to substantially improved inference for a wide range of scenarios. Both LASSO and SPLS (SPLSDA for binary outcomes) provided reasonable results in all simulation scenarios studied, identifying important metabolites without suffering from large numbers of false positives. The only scenario wherein univariate procedures were most reliable was when the sample size was small and the outcome was binary. With a binary outcome, there is relatively little information available to identify associations among a very large number of metabolites; thus, approaches that attempt to model all metabolites at once do not perform as well with smaller sample sizes. Interestingly, we observed the counterintuitive phenomenon that univariate procedures perform worse at identifying significant metabolites as the sample size grows. This appeared to be due to the correlation structure present in the data, which leads to a large number of false positives, and presents a finding with important implications for future analyses of nontargeted metabolomics data.

Human metabolomics studies have historically relied on univariate approaches with an FDR procedure, and less frequently a Bonferroni correction procedure, PCA, or PLSDA without penalization, in the absence of any formal evaluation of optimal statistical methods. While such approaches have proved useful in some respects for analyzing metabolite data, our findings indicate that these approaches may suffer major drawbacks in certain situations. Univariate approaches, as discussed above, can lead to misleading results when the data are intercorrelated, as is nearly always the case in metabolomics studies given common biochemical and biological origins. Other approaches such as PCA or PCR do not directly provide measures of statistical significance, although P values can be generated by permutation testing to completement additional ad hoc measures of variable importance. While metabolomics data may offer some unique challenges, including scale in metabolite levels and missingness across a population, as well as biologically driven intercorrelations, related molecular phenomics fields have similarly suggested that newer statistical approaches may be of great value in identifying statistical significance and prioritization of variables for biological follow up [13,15,16,17,18,19]. Our results suggest that approaches relying on sparsity to perform variable selection lead to quite good performance with respect to all the metrics examined and represent a path forward for future analyses. In particular, when the number of metabolites was similar to or exceeded the number of study subjects, sparse multivariate models exhibited robust statistical power with consistent results, as expected given the design of methods that prioritize sparsity. Thus, nontargeted metabolomics analyses of relatively small cohorts are most likely to benefit from using sparse multivariate models in attempts to identify metabolites associated with a given outcome.

There is, in metabolomics, a strong interest in pathway analyses that might offer some insight regarding the biological mechanisms underlying statistical associations observed between metabolites and a given outcome. These approaches have been established for the vast majority of standard metabolites; for novel metabolites, efforts made towards compound identification can also benefit from pathway mapping, which can offer clues regarding the likely identity of a given unknown molecule [20]. Thus, we elected to use a relatively basic network analysis to visualize inter-metabolite associations and further examine our main findings in this context. We found that univariate approaches tend to identify highly intercorrelated metabolites, whereas multivariate approaches exhibit a more parsimonious as well as broader selection of metabolites. In effect, these findings also support the notion that multivariate compared to univariate approaches tend to select metabolites representing putatively distinct and likely more-orthogonal biological pathways of potential importance and interest in relation to a given outcome. It is important to note, however, that intercorrelated metabolites should not be interpreted as presumably redundant in their potential representation of inter-related biology; in particular, a metabolite in a pathway showing the strongest relationship with a given outcome may be perturbed due to a distinct regulatory enzyme and may be downstream from an original event or upstream from a mechanism causing the greatest clinical effect.

In our study, we found that multivariate approaches that assume some level of sparsity by design (i.e., some metabolites have a very small effect on the outcome) perform with the greatest efficiency for identifying important metabolites. Importantly, this conclusion is based on settings in which the relationship between a metabolite and outcome is linear. In the setting of nonlinearity, it is likely that random forest or other machine learning-based approaches that allow for highly nonlinear relationships may be preferable, although this would require more formal evaluation than provided herein. The value of sparse multivariate analysis may be due to several potential reasons, including the large amount of correlation between metabolites, which requires approaches to examine a metabolite that are conditional to other metabolites. In addition, the fact that many metabolites indeed have little to no association with an outcome of interest favors approaches that enforce sparsity. With these results in mind, we can provide recommendations for future analysis of high dimensional metabolite data. For larger (>1000) samples, multivariate approaches based on sparsity provide a very reasonable strategy to identify important metabolites. In small samples (<200), particularly for binary outcomes, there is no clear-cut ‘best’ method, and the merits of each method depend heavily on the structure of the data. In these cases, utilizing more than one analysis tool in conjunction could help identify key covariates. Importantly, the goal of the study should be taken into account before selecting a statistical approach. If false positives are very undesirable, then we recommend approaches such as LASSO or SPLS that impose sparsity into the model and tend to eliminate presumably less-relevant metabolites.

Our findings can be used to guide the design of future studies, particularly those for which investigators may be interested in estimating a minimum amount of statistical power to detect an association of interest. If pre-existing knowledge of the distribution of metabolite values and their correlation structure is available, either through preliminary or previously analyzed data, then simulation studies similar to those described in this manuscript could be performed. Investigators can use an observed correlation structure to simulate a dataset of metabolite measures and then simulate outcomes given a known range of effect sizes, from which power to detect these effects can be estimated for different statistical approaches.

There are several limitations of the study that merit consideration. The primary findings were based on simulated data, albeit data constructed based on the known structure of an existing high-dimensional dataset derived from actual values in a human cohort. As such, our results may have been influenced by the nature of the underlying artificially created data structure. For this reason, we conducted parallel analyses in a de novo real-world dataset of metabolomics performed in a community-based cohort, and observed results that were largely consistent with those of the simulated data analyses. The observed substantial difference in performance between traditional statistical and statistical learning methods may well have emerged from the difference between univariate and multivariate methods. Accordingly, investigators have suggested that in situations where intercorrelations among predictor variables are expected, a permutation-based FDR approach to univariate analyses should be considered [21]. The extent to which FDR with permutation, or similar variants of univariate analyses, could effectively accommodate correlations and produce different results remains unclear and a subject of ongoing research [6]. The Benjamini–Yekutieli procedure can accommodate varying levels of intercorrelations, and further investigations are needed to evaluate its performance in different data settings [22]. It also should be emphasized that validity of results produced by any statistical model depends not only on model characteristics but also on data quality, which relies on mass spectrometry methods for correctly and consistently identifying metabolites from typical background artefacts [23]. Thus, all statistical analyses are at risk for results of association analyses to be biased to the null due to technical misinterpretations of noise for signal. We did not directly assess potential effects of confounders, including variable batch effects, given that ongoing efforts to develop robust methods are still underway [24]. Importantly, we assigned the assumptions of randomly distributed correlations and non-missingness as fixed attributes of the dataset in order to focus on assessing results from variation in other dataset attributes; given that neither of these assumptions can be made for real data, additional complementary investigations are needed to further understand how various strategies for handling different patterns of intercorrelations and for handling different types of missingness may impact performance for a given modeling method. It is also important to note that this manuscript is focused on evaluating statistical approaches aimed at identifying metabolites associated with a given outcome, wherein such associations could be based not only on mechanisms directly involved in the relative increase or decrease in production of a given metabolite but also on mechanisms that lie upstream to metabolite production (e.g., proximal enzymatic regulators in a system that governs downstream metabolite production). In addition to ascertaining putative mechanistic pathways, another goal in clinical metabolomics research is to identify metabolites most important for predicting a given outcome; to this end, univariate approaches may be equivalent or superior to multivariate approaches, and this is an area that warrants future investigation.

In summary, our findings further indicate that statistical learning approaches aimed at modeling a high-dimensional set of metabolites and their associations with a given outcome warrant more attention in the literature. Taken together, our results suggest that metabolomic analyses should shift towards use of multivariate approaches for identifying distinct markers associated with clinical traits. Univariate approaches, while simple to use, will identify large numbers of false positives when the metabolites are highly correlated with each other—a problem ubiquitous in metabolomics research. If interest lies solely in finding large biologic pathways instead of causal markers (i.e., hypothesis-generating analyses), then univariate approaches may still be useful.

When compared to traditional and frequently employed univariate approaches, statistical learning methods (such as LASSO or SPLS) offer effective and easy to implement options for handling high-dimensional, correlated data of the nature that is commonly seen in metabolomics. In fact, these approaches may well outperform many of the conventionally used methods across a wide variety of scenarios encountered in human metabolomics studies.

## 4. Materials and Methods

### 4.1. Development of Simulated Metabolomics Dataset

We developed a series of simulated metabolomics datasets based on characteristic data features seen in both experimental and human studies (small, case-controlled as well as large cohort) using both targeted and nontargeted mass spectrometry. In particular, we designed the datasets to include the range of structural characteristics typically observed in human plasma metabolomics datasets.

Structural features with respect to outcomes included: (i) binary outcomes in small studies of up to 200 individuals with an outcome frequency of 50%, representing case-control studies with a 1:1 case/control ratio; (ii) binary outcomes in large studies with an outcome frequency of 20% in cohorts with >200 individuals, representing larger observational cohort studies; (iii) continuous outcomes measured in all patients. Structural features with respect to exposures included: (i) number of metabolites ranging from 200 (as is typical of a targeted method) to 2000 (representative of an nontargeted method); (ii) metabolite values following a normal distribution, which is similar to what is commonly observed after a logarithmic or other transformation of the data; (iii) general positive correlation between metabolites, with pairwise correlations randomly distributed around a mean of +0.40 (with random distribution of correlations set as a fixed dataset attribute to allow for a assessing variation in other attributes); (iv) clustering within the data such that large groups of metabolites are highly correlated; and (v) the number of assigned ‘true’ positive metabolites independently associated with the clinical outcome set to 10, with varying effect sizes. The data structures are summarized in Table 1, and an example correlation matrix induced by our simulation design is shown in Figure 5.

### 4.2. Statistical Approaches for Analyzing Metabolomics Data

For comparison of analyses, we applied the following statistical methods to the simulated metabolomics datasets: (1) univariate analyses with multiple testing correction using Bonferroni or false discovery rate (FDR) [28]; (2) principal component regression (PCR) [29,30]; (3) sparse partial least squares (SPLS) [8,10]; (4) sparse partial least squares discriminant analysis (SPLSDA) [8,9]; (5) random forest [7]; and (6) least absolute shrinkage and selection operator (LASSO) [11]. Univariate analyses with multiple testing correction, including Bonferroni correction or Benjamini–Hochberg correction for FDR, have been applied in a variety of, predominantly targeted, metabolomics studies [31,32,33]. Of note, we selected PCR instead of PCA given that PCR has also been applied in prior metabolomics studies and reduces the dimension of the total number of metabolite variables while allowing for feature selection. PCR first reduces the dimensionality of the metabolite data, then uses the selected principal components in a regression model to predict the clinical outcome variable. Importantly, the dimension-reduction step can allow the use of a regression model that otherwise could not be appropriately fit due to the number of metabolite variables exceeding the number of study subjects. Finally, variable importance measures were derived by reallocating the estimated regression coefficients to the metabolites that contributed to each of the chosen principal components. The PLS (partial least squares) regression method maximizes the covariance between a matrix of metabolite variables and the outcome variable, where the outcome is typically a continuous variable; for categorical outcome variables, a variant called PLS discriminant analysis (PLSDA) [34] may be applied. Either PLS or PLSDA serve to decompose metabolite and outcome data into latent structures and aim to maximize the covariance between these latent structures. The random forest method employs a non-parametric ensemble approach to predicting an outcome from metabolomics data by identifying presumably non-linear patterns that may account for metabolite variation in relation to a particular outcome [7]. PCR, PLS, PLSDA, and random forest all suffer from a similar problem when trying to identify important metabolites: While they can rank-order the metabolites in terms of importance, there is no obviously principled way to select a cutoff for which metabolites are ‘significantly’ associated with the outcome. There exist ad hoc approaches to performing variable selection in some of these contexts [35,36]; however, there is no consensus on the appropriate manner for selecting important metabolites. Naïve approaches, such as simply taking the top K covariates to be significant, can be applied, but their properties are not well-understood, and their performance will vary greatly across datasets depending on the true number of significant metabolites present. One way to overcome this issue is to use models that induce sparsity in their respective coefficients. Sparsity refers to the assumption that, adjusting for all measured metabolites, the number of metabolites that are associated with the clinical outcome (‘true’ positives) is far smaller than the overall number of metabolites. These methods differ from dimensionality reduction techniques such as PCA in that the former models include the association with the clinical outcome in the variable-selection process. Accordingly, sparse procedures for PCA and similar models were originally developed to interpret high-dimension data. The most popular of such approaches in the field of statistics is LASSO [11], a method that regresses a given outcome on all metabolite variables simultaneously and achieves parsimonious variable selection by applying a penalty to the magnitude of the regression coefficients. Sparse methods provide automatic variable selection, which solves the aforementioned issue that these methods only allow for variable-importance ranking. Notably, sparse extensions of approaches such as PLS and PLSDA exist [7,8,9,10] and are useful in metabolomics. FDR was implemented using the Benjamini–Hochberg procedure with a false discovery rate of 0.1. Parameters of multivariate approaches such as LASSO, SPLS, and SPLS-DA were selected using cross-validation. Variables were ranked based on the magnitude of the absolute value of the relevant regression coefficients for LASSO, SPLS, and PCR, while variable importance measures were used for random forest. Of note, while univariate and multivariate approaches are often used in combination, we evaluated these methods separately to clarify their distinct potential contributions to what can often exist as a multi-stage data analysis workflow.

It is also important to clarify the distinction between variable selection and significance testing. Methods such as LASSO or other sparse models do not perform significance testing in the traditional sense of controlling type I error or testing hypotheses. Rather, they simply identify a set of metabolites that are relatively important for predicting a given outcome. Thus, herein, we will compare approaches aimed at identifying metabolites of greatest interest in relation to a given outcome, wherein some approaches involve traditional hypothesis testing, and others simply involve variable selection.

To compare the performance between statistical methods in this regard, we evaluated the following metrics: (i) probability of identifying an assigned ‘true’ positive metabolite through variable selection/significance testing as a function of the true effect size (among those methods which allow for such identification); (ii) probability of identifying an assigned ‘true’ positive metabolite as a “top 10” metabolite based on effect size; (iii) average number of false positive metabolites identified by variable selection/significance testing; (iv) positive predictive value (PPV), the probability that a metabolite identified through variable selection/significance testing is truly related to the clinical outcome; (v) negative predictive value (NPV), the probability that a metabolite not identified is truly unrelated to the clinical outcome. These metrics were evaluated separately for continuous and binary outcomes, with all analyses performed using *R*v3.2.3 (R Development Core Team, Vienna, Austria). To visualize the relatedness of metabolites, Spearman correlation coefficients were estimated for all pairs of metabolites, and correlations above +0.75 were isolated, with clusters of these correlations visualized using a D3 visualization framework [37]; within this visualization, metabolites associated with sex and age via Bonferroni, FDR, SPLS, and LASSO were highlighted.

### 4.3. Experimental Human Metabolomics Data

As part of the community-based Framingham Heart Study, the offspring cohort participants underwent a standardized evaluation that included fasting blood sample collection at their eighth examination in 2002–2005, as previously described [38]. All participants provided informed consent and all protocols were approved by the institutional review boards at Boston University Medical Center, Brigham and Women’s Hospital, and the University of California, San Diego. Liquid chromatography–mass spectrometry (LC–MS)-based metabolomics analysis was performed on all available (N = 2895) plasma samples according to previously described protocols [39]. In brief, plasma samples were prepared and analyzed using a Thermo Vanquish ultra-performance liquid chromatographer (UPLC) coupled to a high-resolution Thermo QExactive orbitrap mass spectrometer. Metabolites were isolated from plasma using protein precipitation with organic solvent followed by solid phase extraction. Extracted metabolites underwent chromatographic separation using reverse phase chromatography, whereby samples were loaded onto a Phenomenex Kinetex C18 (1.7 µm, 2.1 × 100 mm) column and eluted using a 7 min linear gradient starting with water:acetonitrile:acetic acid (70:30:0.1) and ending with acetonitrile:isopropanol:acetic acid (50:50:0.02). LC was coupled to a high-resolution Orbitrap mass analyzer with electrospray ionization operating in negative ion mode, with full scan data acquisition across a mass range of 225 to 650 *m/z*. Thermo .raw data files were converted to 32-bit centroid .mzXML using Msconvert (Proteowizard software suite), and resulting .mzXML files were analyzed using Mzmine 2.21, as described in [39]. To eliminate redundant and non-metabolic chromatographic features, we applied the following filters: naturally occurring ^13^C isotopes were consolidated under the monoisotopic peak; common adducts (i.e., H^+^, Na^+^, NH_4_^+^, and K^+^ for positive mode and H^−^, Cl^−^, and acetate for negative mode) were consolidated, with the most-abundant species being reported; multiple charge states were consolidated with the singly charged state, with the most-abundant being reported; and, common ESI in-source fragments (e.g., loss of water) were removed. In addition, all chromatographic features present in a sample blank subjected to the entire sample preparation protocol (with the exception of water being used instead of plasma) were removed. Finally, all remaining chromatographic features were manually inspected for quality in peak shape, retention time consistency, and signal-to-noise ratio, with features exhibiting subpar characteristics subsequently removed. Known compounds that are typically observed in human plasma when applying this method are listed in Appendix A.

From plasma collected from N = 2895 participants, a total of 1933 distinct metabolites were measured with a non-missing value recorded for every participant. We log-transformed and standardized all metabolites to have mean of 0 and standard deviation of 1 due to the expectedly right-skewed nature of the data. Using the same statistical analytical methods described above, we conducted analyses to identify distinct metabolites demonstrating significant associations with age and sex. These phenotypes were specifically selected given both are basic factors available in almost all biomarker analyses, and they allow for analysis of a continuous and binary outcome, respectively.

## 5. Conclusions

We examined the results of both traditional statistical and statistical learning methods across a range of metabolomics datasets. We observed that when the number of metabolites was similar to or exceeded the number of study subjects, as is common with nontargeted metabolomics performed in small cohorts, sparse multivariate models demonstrated the most consistent results and the most statistical power. These findings have important implications for the analysis of metabolomic studies of human disease.

## Figures and Tables

**Figure 1 metabolites-12-00519-f001:**
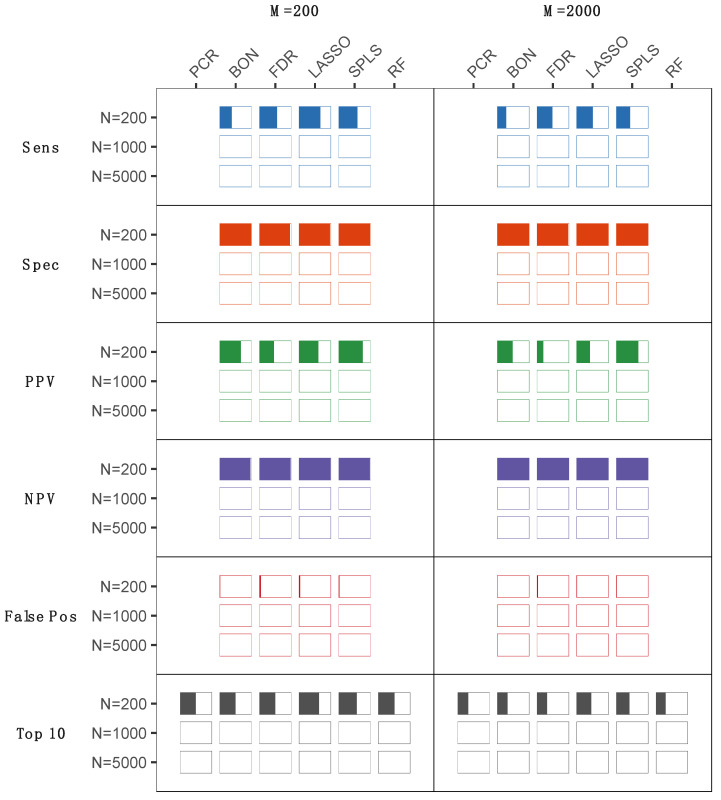
Results for a continuous outcome. The sensitivity, specificity, positive predictive value (PPV), negative predictive value (NPV), and false positive rate are displayed (as percent color fill of each bar) for each statistical method, reflecting their ability to correctly identify the top ten simulated metabolite associations across varying numbers of total metabolite measures (M = 200 or M = 2000) in study samples collected from varying numbers of study subjects (N = 200, N = 1000, or N = 5000). PCR, principal components regression; BON, Bonferroni; FDR, false discovery rate; LASSO, least absolute shrinkage and selection operator; SPLS, sparse partial least squares; RF, random forest.

**Figure 2 metabolites-12-00519-f002:**
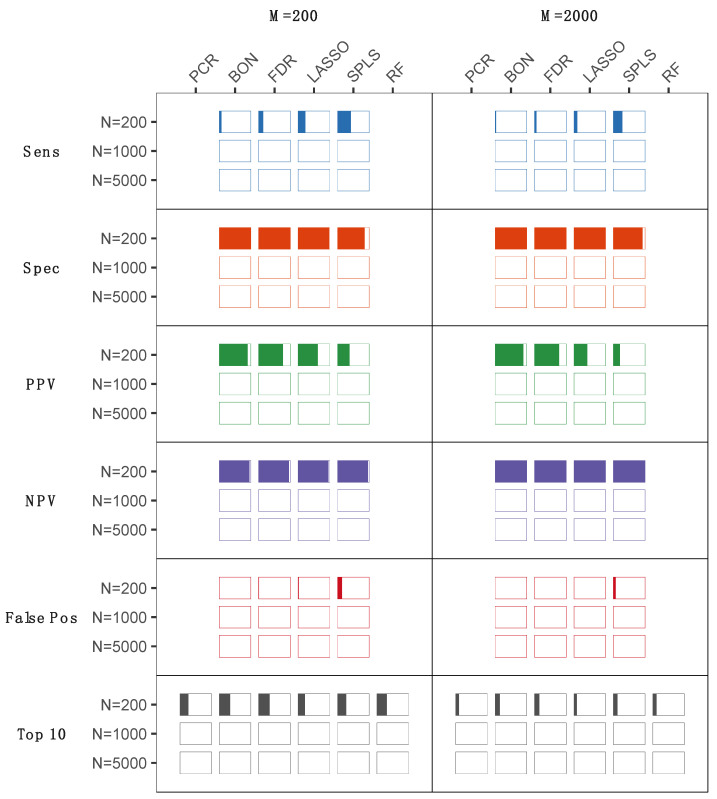
Results for a binary outcome. The sensitivity, specificity, positive predictive value (PPV), negative predictive value (PPV), and false positive rate are displayed (as percent color fill of each bar) for each statistical method, reflecting their ability to correctly identify the top ten simulated metabolite associations across varying numbers of total metabolite measures (M = 200 or M = 2000) in study samples collected from varying numbers of subjects (N = 200, N = 1000, or N = 5000). PCR, principal components regression; BON, Bonferroni; FDR, false discovery rate; LASSO, lease absolute shrinkage and selection operator; SPLS, sparse partial least squares; RF, random forests.

**Figure 3 metabolites-12-00519-f003:**
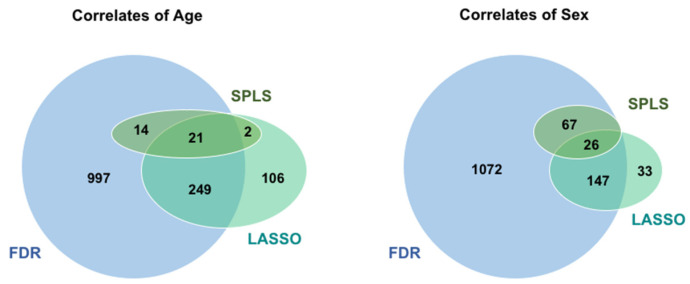
Analysis of actual, experimentally derived metabolomics data. The number of metabolites found in association with age (continuous outcome) and sex (binary outcome) from experimentally derived metabolomics studies (see text) for different statistical methods: false discovery rate (FDR), sparse partial least squares (SPLS), and least absolute shrinkage and selection operator (LASSO). The number of metabolite correlates found in common by the different methods is relatively small compared to the total number of apparently significantly associated metabolites.

**Figure 4 metabolites-12-00519-f004:**
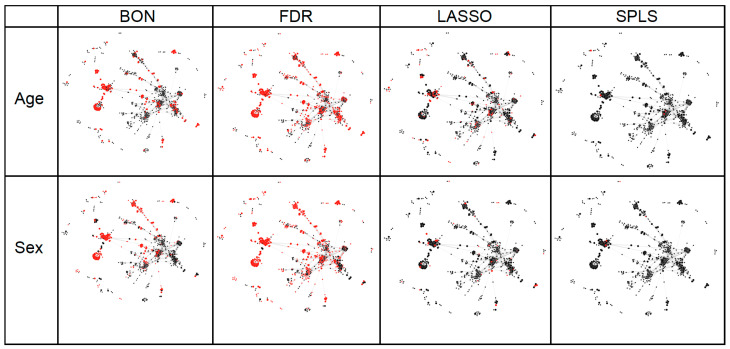
Putative network distribution of metabolites identified by different methods used to analyze cohort-based metabolomics data. The number of metabolites found in association with age (continuous outcome) and sex (binary outcome) from experimentally derived metabolomics studies (see text) for the different statistical methods applied was greater for traditional than for statistical learning models. Notably, the former identified metabolites that tended to be highly correlated with each other (Spearman rho ≥ 0.65), whereas the latter identified a more parsimonious number of metabolites distributed across the putative network of all highly intercorrelated metabolites. BON, Bonferroni; FDR, false discovery rate; LASSO, least absolute shrinkage and selection operator; SPLS, sparse partial least squares.

**Figure 5 metabolites-12-00519-f005:**
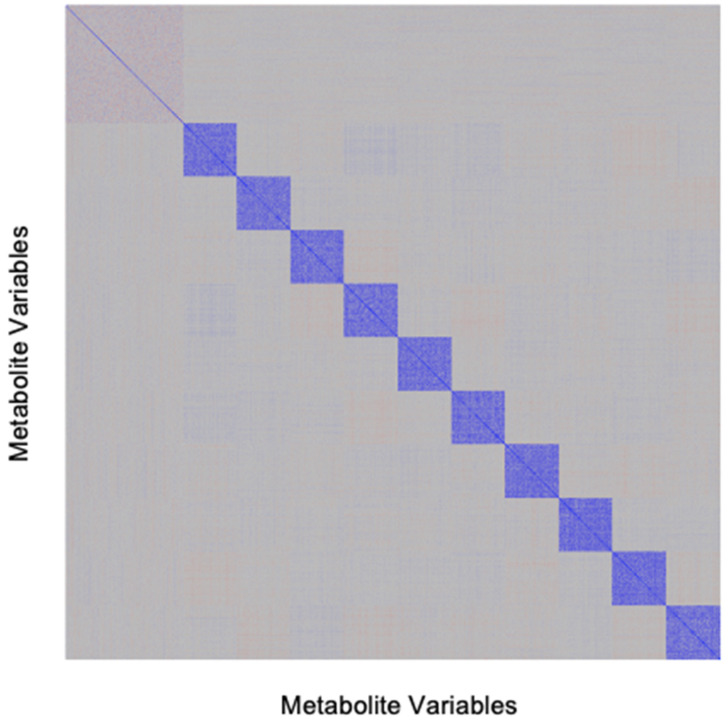
Structure of the simulated dataset. To perform statistical analyses within a controlled environment with pre-specified metabolite–outcome associations, we created a simulated dataset based generally on data features observed in multiple real-world datasets. One such simulated dataset demonstrates a scenario with multiple clusters of metabolites that have within-cluster correlation but little cross-cluster, mimicking the inter-relationships observed in actual experimentally derived human metabolomics studies [25,26,27].

**Table 1 metabolites-12-00519-t001:** Data structures used for analyses.

Dataset	Outcome Characteristics	No. of Metabolites	No. of Observations (i.e., No. of Persons)
1	Continuous	200	200
2	Continuous	200	1000
3	Continuous	200	5000
4	Continuous	2000	200
5	Continuous	2000	1000
6	Continuous	2000	5000
7	Binary: 20% frequency	200	200
8	Binary: 50% frequency	200	1000
9	Binary: 50% frequency	200	5000
10	Binary: 20% frequency	2000	200
11	Binary: 50% frequency	2000	1000
12	Binary: 50% frequency	2000	5000

## Data Availability

The datasets generated and/or analyzed during the current study are available in the dbGap repository: Framingham Cohort dbGaP Study Accession. Available online: https://www.ncbi.nlm.nih.gov/projects/gap/cgi-bin/study.cgi?study_id=phs000007.v29.p10 (accessed on 3 February 2022).

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
