# Peer review of "Quantitative Comparison of Statistical Methods for Analyzing Human Metabolomics Data"

_metabolites, 2022, doi:10.3390/metabo12060519_

Round 1

Reviewer 1 Report

The authors show a comprehensive analysis of which statistial analysis is best used for metabolomic datasets with different n sizes and how the results differ depending on that.

The study is nicely laid out and investigates important points that are important for other researchers to read and know about. The only point I would like to see addressed is to have the same analysis re-done for a dataset with n=30 or n=50, as these numbers are the ones I most often see in our lab. Studies with n>=200 are not that common. 

Reviewer 2 Report

The authors have conducted a study using both simulated and real metabolomic datasets to evaluate which statistical methods work best on datasets. The study is interesting and I think will be of interest to the readers of Metabolites. I have some minor concerns about both the study itself and the way it was reported.

I could not evaluate the supplementary material because this was not provided, and the datasets referred to at the bottom of the article led to a dead link.

General points

Firstly, the authors are concentrated on finding “true positives”. Their definition of a true positive does not truly fit the statistical definition of this term, and seemed to concentrate on limiting correlated metabolites from the analysis (which statistically, may also have been true positives). As such, they rather glossed over the potential uses that can be made in interpretation when correlated metabolites are found and can be clustered in single or related metabolic pathways. Secondly, they have not considered that the metabolite in a pathway showing the strongest relationship to a particular outcome may be perturbed due to a particular regulatory enzyme and may be downstream of the original event and/or upstream of the mechanism causing the most damage.

Secondly, their methods section describing their simulated datasets was a little unclear and left me wondering how it was constructed, the number of repetitions they had for any individual modelling situation (was only one dataset used, or were different datasets modelled for all of the stated conditions) and the range of variability across the dataset in both effect size and distribution i.e. it is common to see some strange bi or multimodel distributions of individual metabolites. I would have liked to have seen more common approaches such as PCA being tested and no mention was made of the various validation parameters in play to validate more sophisticated models. In addition, little thought had been given to missing values in the dataset and how this affected the results – especially when poor detection of a metabolite A is combined with good detection of a correlated metabolite B, or where a metabolite C has increasingly poor detection over the course of a run. Similarly unrelated confounding factors such as batch effects were not discussed. Lastly, it is a common approach to combine multivariate and univariate approaches and this was not tried at all by the authors which is surprising given the frequency of this approach. Their conclusions seem based purely on the number of variables to number of subjects ratio and is perhaps over confident for this reason. I would strongly recommend they discuss these issues clearly in the discussion and modify the strength of their conclusions in both the abstract and the conclusions part of the paper.

Detailed points

Line 51: The original definition of metabolomics was a global analysis of all metabolites in a cell or biological system. Metabolomics started as non targeted, moved into targeted and is increasingly going back to non targeted. I find the history given here of the field is misleading and should be changed to more accurately reflect the development of metabolomics, or deleted entirely.

 Line 71: While FDR is used quite often, Bonferroni is used very rarely in metabolomics and is more applied to genomics. It is inappropriately strict for metabolomics precisely because its basic premise is independent variables, which metabolites are not.

Line 269: a p value can be calculated for most multivariate models based on permutation testing.

Line 286: the number of truly novel metabolites found is small, whereas the biochemical pathways are known for the vast majority of standard metabolites. While compound identification can sometimes still be an issue, one of the advantages of pathway mapping is that it may give clues as to the likely identity of the unknown.

Line 314: this goes against centuries of statistical advice which states you should choose your statistical test before analysing the data. Indeed, it is considered bad practice by most statisticians to adopt the approach suggested here.

 Line 348: I think the authors may have forgotten a basic tenet of biochemistry here, which is that there are certain enzymes that act as regulators in a system. Where metabolites build up due to these enzymes being the bottle neck, the actual mechanism of action may be occurring further upstream.

Line 385 (iii): Correlations in real data are only very rarely randomly distributed. Justify why a random correlation approach was used here.

And (iv) even when metabolites are part of the same pathway, this does not always mean they are well correlated, especially where one metabolite is present in multiple pathways, or its function and regulation are based on its distribution in a cell, or bound state in plasma.

Line 404: Provide some evidence for this statement

Line 412: there is a version of the BH correction which allows for variables to be correlated. Why was this not tried?

Line 512: How were missing values dealt with in this dataset.

Reviewer 3 Report

In the manuscript “Quantitative Comparison of Statistical Methods for Analyzing Human Metabolomics Data” the authors Henglin et al. addresses a hot and interesting topic in the field of metabolomics and mass spectrometry. The aim of this study was to compare and evaluate multiple statistical approaches for the analysis of omics data obtained from human plasma samples.

In my opinion the manuscript is very well written and of high value to the readers.  

Round 2

Reviewer 1 Report

I stand by my first review, that this paper is highly relevant but I think the authors missed a nice opportunity to showcase small datasets as well. 

Author Response

We thank the Editor and Reviewers for their insightful suggestions, which have considerably improved the manuscript. We have sought to address each of the issues in the following point-by-point response.

Response:  We very much appreciate the kind comments from the Reviewer. We completely agree that studies examining the effect of variation in sample size would be of great interest. To allow for fair comparisons across the range of sample sizes, we have now completely re-run all analyses across most of the originally studied sample sizes including smaller datasets of N=50 and N=100. Given that the originally studied maximum sample size was N=5000 (which requires several weeks of computational time to re-run), we have completed re-analyses including sample sizes from N=50 through N=1000 (which allows for the re-run to be completed in a shorter period of time). Based on our careful review of all analysis output generated to date, we anticipate that a complete re-run including samples sizes of N=5000 or greater will yield very similar results to those including sample sizes up to N=1000. Thus, the results of the analyses re-run to date, including smaller sample sizes starting at N=50 and up to larger samples sizes up to N=1000, are now displayed below. These results show generally similar findings to the originally reported results, with respect to decreasing sensitivity for progressively smaller samples sizes albeit with specificity and negative predictive value relatively preserved. Notably, we did find that the positive predictive value was reduced and, relatedly, the false positive rate was increased when SPLS was applied to the smaller sized datasets (i.e. N=50 and N=100) and so we have now added report of these findings to the revised manuscript. We thank the Reviewer for requesting these informative additional analyses, which we believe has strengthened the overall manuscript.

Page 3, Line 147: “In secondary analyses, we repeated the simulation experiments to include smaller samples sizes (i.e. N=50 and N=100) that may be used for some studies despite acknowledged limitations in statistical power. In these analyses, we observed the expected trend of generally reduced sensitivity for smaller samples sizes while specificity was generally preserved, corresponding to a greater positive predictive value (PPV) in smaller sample sizes. The exception was SPLS models, for which false positive rates were increased in the smallest samples sizes along, corresponding to reduced PPV (Supplementary Figure 1).”

Page 6, Line 188: “An exception to this trend was seen in the smallest-sizes samples (i.e. N=50 or N=100) wherein the false positive rate was higher for SPLS than for LASSO models.”

Reviewer 2 Report

The authors have made substantial efforts to address my concerns and I have now only minor points for correction:

The authors use some terms in their manuscript such as false positive; can they confirm that they are using the generally accepted definition of this term and add this definition somewhere in the manuscript.

In abstract Line 34: „higher false discovery rate due to substantial correlations among metabolites“ Correlations among metabolites do not necessarily make the discovery „false“ in a statistical sense. It may just make it less biologically informative. Please reword or justify the statement.

Line 740: „Because many statistical methods are unable to simultaneously model a number of metabolites which exceeds the number of study subjects”

Indeed, PCA was developed to allow exactly this….Sparse models of PCA or other models were originally developed to enable better interpretation of the data. Please change.

Author Response

  1. The authors use some terms in their manuscript such as false positive; can they confirm that they are using the generally accepted definition of this term and add this definition somewhere in the manuscript.

Reply: We thank the Reviewer for pointing out the very important need to clarify use of terms such as ‘false positive’. We can confirm that we use this particular term to refer to the occurrence of an error in the binary classification context, specifically the instance wherein a ‘true negative’ association is erroneously classified as a positive association. In our simulation experiments, given that we pre-specified a set of certain metabolites as having true positive associations (i.e. directly associated with the simulated clinical outcome), then any metabolite selected by a model (i.e. identified as being associated with the outcome) that is not a member of the pre-specified set of ‘true positives’ is ultimately considered a ‘false positive’ identified by the model. As helpfully advised by the Reviewer, we have now clarified this definition in the manuscript:

            Page 5, Line 177: “…false positives (i.e. variables identified as significantly associated but not pre-specified as directly associated with the clinical outcome) increased substantially with univariate methods, in the setting of these variables being selected due to their correlation with ‘true positive’ metabolites (i.e. variables pre-specified as directly associated with the clinical outcome)…”

  1. In abstract Line 34: „higher false discovery rate due to substantial correlations among metabolites“ Correlations among metabolites do not necessarily make the discovery „false“ in a statistical sense. It may just make it less biologically informative. Please reword or justify the statement.

Response: We completely agree with this important point made by the Reviewer. We have now revised this section of text in the manuscript, as helpfully suggested:

            Page 1, Line 34:  “…univariate compared to multivariate methods resulted in an apparently higher false discovery rate in the context of substantial correlations between metabolites directly associated with the outcome and metabolites not associated with the outcome. Although the higher frequency of such associations would not be considered false in the strict statistical sense, they may be considered biologically less informative.”

  1. Line 740: „Because many statistical methods are unable to simultaneously model a number of metabolites which exceeds the number of study subjects” Indeed, PCA was developed to allow exactly this….Sparse models of PCA or other models were originally developed to enable better interpretation of the data. Please change.

Response: We again completely agree with this astute comment from the Reviewer. We have now revised this section of the manuscript, as very helpfully suggested:

            Page 13, Line 631: “PCR first reduces the dimensionality of the metabolite data, then uses the selected principal components in a regression model to predict the clinical outcome variable. Importantly, the dimension reduction step can allow the use of a regression model which otherwise could not be appropriately fit in the context of the number of metabolite variables exceeding the number of study subjects.”

            Page 14, Line 747: “These methods differ from dimensionality reduction techniques such as PCA in that the former models include the association with the clinical outcome in the variable selection process. Accordingly, sparse procedures for PCA and similar models were originally developed to facilitate interpretation of high-dimensional data.”